# IRCAN: Mitigating Knowledge Conflicts in LLM Generation via Identifying and Reweighting Context-Aware Neurons

**Dan Shi, Renren Jin, Tianhao Shen, Weilong Dong, Xinwei Wu, Deyi Xiong**[*]
College of Intelligence and Computing, Tianjin University, Tianjin, China
{shidan, rrjin, thshen, willowd, wuxw2021, dyxiong}@tju.edu.cn

## Abstract

It is widely acknowledged that large language models (LLMs) encode a vast reservoir of knowledge after being trained on mass data. Recent studies disclose knowledge conflicts in LLM generation, wherein outdated or incorrect parametric knowledge (i.e., encoded knowledge) contradicts new knowledge provided in the context. To mitigate such knowledge conflicts, we propose a novel framework, IRCAN (Identifying and Reweighting Context-Aware Neurons) to capitalize on neurons that are crucial in processing contextual cues. Specifically, IRCAN first identifies neurons that significantly contribute to context processing, utilizing a context-aware attribution score derived from integrated gradients. Subsequently, the identified context-aware neurons are strengthened via reweighting. In doing so, we steer LLMs to generate context-sensitive outputs with respect to the new knowledge provided in the context. Extensive experiments conducted across a variety of models and tasks demonstrate that IRCAN not only achieves remarkable improvements in handling knowledge conflicts but also offers a scalable, plug-and-play solution that can be integrated seamlessly with existing models. Our codes are released at `https://github.com/danshi777/IRCAN`.

## 1 Introduction

Large language models (LLMs), trained on extensive data, are known for encapsulating a broad spectrum of knowledge [13, 40, 26, 35]. However, due to the rapid evolution of information/knowledge as well as noise in training data, LLMs may possess incorrect or outdated knowledge. To mitigate this issue, in real-world applications, methods like retrieval-augmented generation (RAG) are usually used to integrate latest event knowledge or knowledge from external databases into the context of prompts fed into LLMs. This enables online updates to knowledge and the incorporation of domain-specific information, enhancing the accuracy and reliability of the outputs of LLMs.

Such generation formalism equips LLMs with two sources of knowledge: (1) *parametric knowledge*, which is acquired during pre-training and encoded within model parameters; and (2) *contextual knowledge*, which is supplied as the prefix context within the input [3]. However, previous studies have shown that when LLMs encounter contradictions between these two types of knowledge (known as *knowledge conflicts* [43, 44]), they may overly adhere to their inherent parametric knowledge and fail to pay sufficient attention to new knowledge introduced in the context [28, 4, 43], leading to hallucinations [48, 19, 18]. For example, although we present the latest information "As of 2023, India has surpassed China as the most populous country." in the context to LLaMA-2-7B, when it is faced with the question "Which country is the most populous in the world?\nAnswer:", it still provides the outdated answer "China".

---

[*]Corresponding author

We hypothesize that within LLMs, there exist neurons that specifically focus on processing context, akin to knowledge neurons [8]. With this assumption, to alleviate the aforementioned issues, we propose a framework IRCAN for **I**dentifying and **R**eweighting **C**ontext-**A**ware **N**eurons to encourage the model to pay more attention to contextual knowledge during generation. Specifically, we first measure the contribution of each neuron to the context processing by calculating their attribution scores. Subsequently, we increase the weights of the detected context-aware neurons, which allows the model to effectively up-weight the contextual knowledge during generation.

We conduct extensive experiments on a diverse array of models from multiple families, including LLaMA [39], Gemma [31] and Amber [27], spanning various parameter scales (2B, 7B, 8B, 13B) and encompassing both pre-trained and instruction-tuned models. To conduct a comprehensive evaluation, we carry out experiments on two types of tasks: completion and multiple-choice. Experiment results demonstrate that our method can effectively identify neurons responsible for processing the context within LLMs. Moreover, by enhancing these neurons, LLMs can be guided to remain more faithful to the information provided in the context when facing knowledge conflicts, rather than sticking to its intrinsic knowledge. Additionally, our method can serve as a plug-and-play module, easily integrated with existing approaches. In completion tasks, IRCAN has achieved state-of-the-art performance, with substantial improvements of 129% and 136% in terms of accuracy for LLaMA-2-7B and LLaMA-3-8B respectively. Remarkably, when our method is integrated with CAD [36], a previous strong method for dealing with knowledge conflicts, the performance of LLMs can be further improved. In multiple-choice tasks, IRCAN achieves comparable results to the baseline, and when combined with CAD, our method sets new state-of-the-art results.

The main contributions of our work are summarized as follows:

- We pioneer the exploration of attribution methods to knowledge conflicts for LLMs, offering a novel approach to resolving knowledge conflicts.
- We propose an attribution method to identify neurons within LLMs that are responsible for processing context based on integrated gradient. Furthermore, by enhancing these context-aware neurons, the LLMs' fidelity to contextual knowledge is effectively improved.
- We conduct extensive experiments and experiment results demonstrate that the proposed approach can significantly boost the performance of LLMs on tasks involving knowledge conflicts.

## 2 Related Work

To correct outdated or incorrect knowledge in language models, previous studies have explored three main strategies: fine-tuning, model editing and contrastive decoding.

**Fine-tuning**   Fine-tuning aims to update the internal knowledge of an existing LLM through further training on additional data, including datasets with the latest information or domain-specific datasets [50, 20, 11, 45, 17]. However, this process requires substantial computational resources and a large amount of training data, as well as significant training time, which can be unaffordable in many cases. More seriously, it may lead to catastrophic forgetting issues.

**Model Editing**   Model Editing seeks to edit incorrect or undesirable knowledge encoded in pre-trained models. Some studies initially identify knowledge-related parameters of the existing pre-trained models and then directly edit particular knowledge into these parameters [12, 29, 30, 41, 42]. Other efforts have been explored to store new or correct knowledge in an extra memory, replacing the original predictions with this knowledge during generation [33, 9]. Additionally, meta-learning based methods learn to edit models through meta-learning [2, 32]. However, these approaches are only applicable to modifying specific knowledge. In contrast, our method is independent of specific knowledge: regardless of the type of knowledge contained in the context, it enhances the LLM's utilization of this knowledge.

**Contrastive Decoding**   Contrastive decoding strategies are adopted during generation, which amplify the differences in output probabilities between various model scales [23] or different layers of an LLM [5], thereby reducing hallucinations. Among these, context-aware decoding (CAD) [36] amplifies the difference between output probabilities with and without context, encouraging the LLM

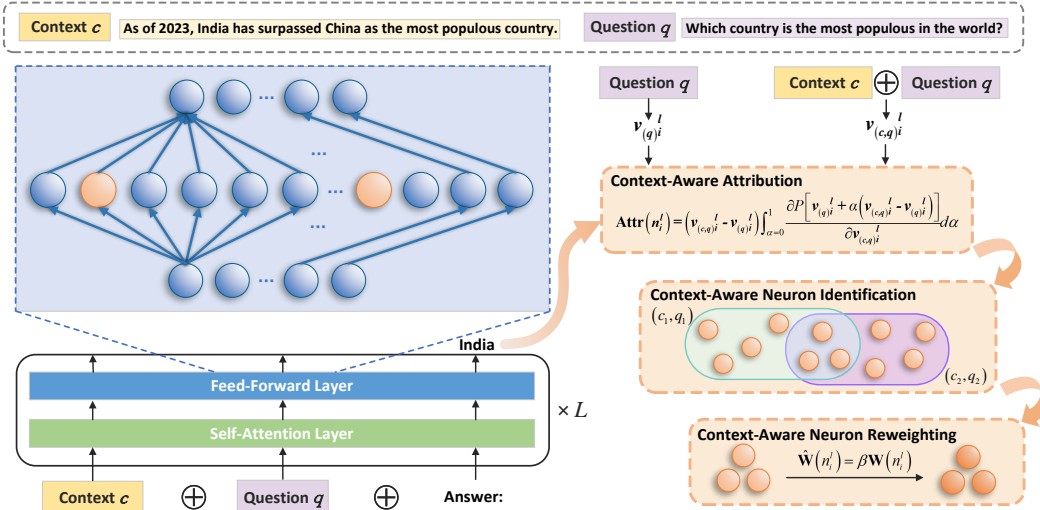

Figure 1: The diagram of IRCAN. When an LLM faces a knowledge conflict between the context and its inherent knowledge, IRCAN first calculates the attribution score for each neuron to measure its contribution to processing the context. It then identifies context-aware neurons by taking the intersection of neurons with the highest scores. Subsequently, the identified neurons are reweighted so that IRCAN could guide the model to be more aligned with the contextual knowledge, ensuring greater fidelity to the context.

to attend to its context during generation. Since its task and goal are the same as ours, we utilize it as a baseline for comparison in our experiments. Significantly different from CAD, our approach IRCAN operates at a finer granularity of neurons, thereby providing a degree of interpretability for analyzing and resolving knowledge conflict issues.

## 3 Methodology

We focus on tasks involving context-specific knowledge conflicts. The input of these tasks is formulated as $(c, q)$, where $c$ is the context, and $q$ represents the question in completion tasks or the question combined with a suffix consisting of choices in multiple-choice tasks. We propose a novel method that is dedicated to improving the faithfulness of LLMs to the context to address these tasks. The proposed IRCAN methodology is structured into three phases: Initially, we compute the attribution scores of each neuron to assess its influence on context processing. Subsequently, neurons that are responsive to context, termed "context-aware neurons", are identified. In the final step, we enhance the influence of these detected neurons through a reweighting process, thereby augmenting their impact on the model's generation. The framework of IRCAN is illustrated in Figure 1.

### 3.1 Context-Aware Attribution

Previous work has found the existence of knowledge neurons that store and express factual knowledge in FFNs [8]. We speculate that certain neurons responsible for processing contextual knowledge also exist in FFNs. Inspired by Hao et al. [14], who introduce an attribution method to interpret the information interactions inside Transformer, we propose a context-aware attribution method based on integrated gradients [37] to identify these neurons. Our method calculates the contribution scores of FFN neurons in perceiving the context towards predicting answers. This evaluation helps determine which neurons play a critical role in context processing.

The attribution score of each neuron to be evaluated is denoted as $\text{Attr}(n_i^l)$, where $n_i^l$ represents the intermediate neuron at the $i$-th position in the $l$-th FFN layer of the language model. Initially, we take only the question as input, record the activation value of each neuron and denote it as $\boldsymbol{v}_{q_i}^l$. Subsequently, we input both the context and the question into the language model and record the new activation value, denoted as $\boldsymbol{v}_{(c,q)_i}^l$. To calculate the attribution score $\text{Attr}(n_i^l)$, we gradually change

the activation value of a neuron $n_i^l$ from $\boldsymbol{v}_{q_i}^l$ to $\boldsymbol{v}_{(c,q)_i}^l$ when the input consists of both context and question. At the same time, the output probability of the model changes accordingly. We calculate the probability of the correct answer predicted by the language model, denoted as:

$$P(\boldsymbol{v}_i^l) = p(y^*|c, q, \mathbf{A}(n_i^l) = \boldsymbol{v}_i^l), \tag{1}$$

where $y^*$ denotes the correct answer; $\boldsymbol{v}_i^l$ is a given value assigned to the neuron activation $\mathbf{A}(n_i^l)$. We integrate the gradient of the probability during this process as the neuron's context-aware attribution score, as follows:

$$\text{Attr}(n_i^l) = \left(\boldsymbol{v}_{(c,q)_i}^l - \boldsymbol{v}_{q_i}^l\right) \int_{\alpha=0}^{1} \frac{\partial P\left[\boldsymbol{v}_{q_i}^l + \alpha \left(\boldsymbol{v}_{(c,q)_i}^l - \boldsymbol{v}_{q_i}^l\right)\right]}{\partial \boldsymbol{v}_{(c,q)_i}^l} \, d\alpha, \tag{2}$$

where $\frac{\partial P\left[\boldsymbol{v}_{q_i}^l + \alpha\left(\boldsymbol{v}_{(c,q)_i}^l - \boldsymbol{v}_{q_i}^l\right)\right]}{\partial \boldsymbol{v}_{(c,q)_i}^l}$ calculates the gradient of the model probability with regard to $\boldsymbol{v}_{(c,q)_i}^l$, $\alpha$ controls the integration from $\boldsymbol{v}_{q_i}^l$ to $\boldsymbol{v}_{(c,q)_i}^l$.

Theoretically, the integrated gradients technique adheres to two fundamental axioms of attribution methods: *Sensitivity* and *Implementation Invariance* [37]. The *Sensitivity* axiom stipulates that if modifying a neuron alters the prediction, that neuron should be assigned a non-zero attribution score. The *Implementation Invariance* axiom dictates that the attributions should remain identical for two networks with equivalent functionality. Adherence to these axioms ensures that the attribution scores accurately reflect the importance of neurons and are invariant to implementation details. The attribution scores facilitate the identification of neurons essential for context processing.

Intuitively, by integrating over the gradient as $\alpha$ changes from 0 to 1, $\text{Attr}(n_i^l)$ accumulates the output probability changes caused by the activation value changes from the absence to the presence of context. If the neuron has a strong perception and processing capability regarding the context, the gradient will be significant, resulting in a large integration value. Therefore, the attribution score can measure the neuron's sensitivity to the context and its contribution to processing the context.

Directly calculating continuous integrals is intractable. We instead use the Riemann approximation of the integration to efficiently compute the attribution score. Specifically, we sum the gradients at points occurring at sufficiently small intervals from $\boldsymbol{v}_{q_i}^l$ to $\boldsymbol{v}_{(c,q)_i}^l$:

$$\tilde{\text{Attr}}(n_i^l) = \frac{\left(\boldsymbol{v}_{(c,q)_i}^l - \boldsymbol{v}_{q_i}^l\right)}{m} \sum_{k=1}^{m} \frac{\partial P\left[\boldsymbol{v}_{q_i}^l + \frac{k}{m}\left(\boldsymbol{v}_{(c,q)_i}^l - \boldsymbol{v}_{q_i}^l\right)\right]}{\partial \boldsymbol{v}_{(c,q)_i}^l}, \tag{3}$$

where $m$ is the number of approximation steps. Following previous work [8], we set $m$ to 20, which performs well in our experiments.

## 3.2 Context-Aware Neuron Identification

Based on the calculated neuron attribution scores $\text{Attr}(n_i^l)$, we first retain neurons with scores above the threshold $t$, creating a coarse set of context-aware neurons. Then, for each example, we select the top $z$ neurons with the highest attribution scores as the candidate set. In our experiments, $t$ and $z$ are set to 10% and 20 respectively. Ultimately, we count the number of co-occurrences of neurons in all candidate sets, and we select the top $h$ neurons with the highest number of co-occurrences as context-aware neurons. These identified context-aware neurons are shared across all data instances.

## 3.3 Context-Aware Neuron Reweighting

In order to make LLMs generate outputs that are more faithful to the context, we enhance the identified context-aware neurons. We adopt a simple yet effective enhancement measure:

$$\hat{\boldsymbol{W}}(n_i^l) = \beta \boldsymbol{W}(n_i^l), \tag{4}$$

i.e., amplifying the weights of these neurons to $\beta$ (i.e., enhancement strength) times their original weights. This adjustment amplifies the role these neurons play as information flows through them, thus enhancing their influence on the model's output.

# 4    Experiments

We conducted experiments on two different types of knowledge conflict tasks (i.e., completion and multiple-choice) to verify the effectiveness of IRCAN. These tasks demand the model to reason about information in the context and generate context-faithful responses.

## 4.1    Dataset

**Completion Task**    We conducted completion task experiments on the MemoTrap [25] dataset. It evaluates the models' ability to adhere to the given context in order to generate an unfamiliar phrase, rather than defaulting to a well-known phrase that has been encoded in its parameters during training. Specifically, it consists of instructions that challenge the language model to conclude a well-known proverb with a terminal word that diverges from its traditional ending (e.g., "Write a quote that ends in the word 'returned': Long absent, soon ____", where the commonly used ending is "forgotten"). It is designed to explore the potential for language models to fall into memorization traps.

**Multiple-choice Task**    For the multiple-choice task, we utilized the COSE_KRE and ECARE_KRE datasets [46], which were derived from ECQA [1] and e-CARE [10], respectively. The derivation process involved selecting one of the incorrect answer choices and prompting ChatGPT to generate explanations supporting this incorrect answer. Specifically, the selected incorrect answer was treated as the correct answer, and the explanations generated by ChatGPT were used as the context for the multiple-choice question.

Illustrative examples from all datasets are shown in Table 4 in Appendix A. We expect LLMs to pay more attention to the knowledge in the context.

## 4.2    Models and Metrics

For the completion task, we conducted experiments on four LLMs with diverse parameter scales: Gemma-2B [31], Amber [27], LLaMA-2-7B and LLaMA-2-13B [39]. We employed accuracy as the evaluation metric, which quantifies the proportion of correctly generated words. The prompt was formed by combining the context and question, allowing the LLMs to generate a continuation. Regular expressions were used to extract the generated ending word.

For the multiple-choice task, we evaluated a diverse list of LLMs of the chat version: LLaMA-2-7B-Chat, LLaMA-2-13B-Chat, LLaMA-3-8B-Instruct[2] and Gemma-2B-it [31]. We evaluated the performance of these LLMs by measuring the accuracy of selecting the correct answer. We perform prompt engineering to design the appropriate prompt for each model. Please refer to Appendix B for more details. We selected the answer based solely on the highest probability among options, which is the official implementation of MMLU [15] and widely used in other benchmarks [16, 21].

Although we used accuracy as the primary metric to evaluate our method more comprehensively, we also designed a supplementary metric: stubbornness rate (SR), which measures whether the LLM persistently adheres to its internal memorized knowledge. This metric is defined as the accuracy with which the model's generation matches the original golden label (for the MemoTrap dataset, i.e., the common ending word of a well-known proverb; for the COSE_KRE and ECARE_KRE dataset, i.e., the original golden option). A lower stubbornness rate indicates that the LLM exhibits a decreased reliance on the knowledge encapsulated within its internal parameters during the generation process, suggesting a greater propensity to incorporate contextual information to a certain extent.

## 4.3    Baselines

To demonstrate the effectiveness of IRCAN, we compared it with the following baselines: **Original**: which refers to the LLMs without any modification. **Prompt engineering based methods**: we curated three types of prompts to explicitly instruct LLMs to pay more attention to the knowledge in context on the multiple-choice task. According to the content added in the prompt, we denote these three prompt engineering based methods as Based_on, Based_on_Formatted, and Utilizing_Formatted, respectively. Please refer to Appendix C for the details of these prompts. **Inference-Time Intervention (ITI)** [22]:

---

[2]https://llama.meta.com/llama3/

| Models | Gemma-2B | | LLaMA-2-7B | | Amber (7B) | | LLaMA-3-8B | | LLaMA-2-13B | |
|---|---|---|---|---|---|---|---|---|---|---|
| | ACC ↑ | SR ↓ | ACC ↑ | SR ↓ | ACC ↑ | SR ↓ | ACC ↑ | SR ↓ | ACC ↑ | SR ↓ |
| Original | 23.24 | 35.82 | 24.52 | 50.96 | 24.95 | 48.40 | 20.26 | 53.30 | 27.08 | 46.70 |
| ITI (Probe Weight Direction) | 26.01 | 25.16 | 31.77 | 44.78 | 20.26 | 43.50 | 18.34 | 53.52 | 23.03 | 51.17 |
| ITI (Mass Mean Shift) | 0.00 | 0.00 | 31.34 | 44.99 | 0.00 | 0.00 | 18.12 | 53.94 | 22.60 | 52.45 |
| CAD | 24.52 | 21.96 | 44.56 | 32.84 | 36.07 | 34.97 | 39.66 | 36.03 | 39.23 | 23.24 |
| IRCAN | 24.73 | 30.28 | 56.08 | 18.55 | 41.15 | 31.56 | 47.76 | 20.68 | 52.24 | 14.29 |
| IRCAN-CAD | **27.08** | **17.27** | **61.83** | **12.79** | **45.84** | **25.59** | **54.37** | **16.84** | **58.64** | **9.38** |

Table 1: Results (in %) of the completion task on the MemoTrap dataset. The best results are highlighted in **bold**, and the second-best results are underlined.

| Datasets | Models | Gemma-2B-it | | LLaMA-2-7B-Chat | | LLaMA-3-8B-Instruct | | LLaMA-2-13B-Chat | |
|---|---|---|---|---|---|---|---|---|---|
| | | ACC ↑ | SR ↓ | ACC ↑ | SR ↓ | ACC ↑ | SR ↓ | ACC ↑ | SR ↓ |
| COSE_KRE | Original | 35.02 | 21.28 | 36.66 | 23.40 | 39.93 | 47.79 | 49.75 | 29.13 |
| | Based_on | 34.70 | 22.42 | 33.22 | 20.29 | 42.88 | 45.34 | 50.57 | 29.46 |
| | Based_on_Formatted | 38.46 | 22.42 | 32.41 | 18.49 | 51.55 | 37.81 | 41.24 | 23.57 |
| | Utilizing_Formatted | 38.95 | 22.26 | 33.06 | 18.00 | 50.08 | 40.10 | 41.57 | 21.93 |
| | Opin | 35.19 | 19.97 | 35.19 | **17.35** | **60.23** | **30.11** | 43.21 | 22.91 |
| | ITI (Probe Weight Direction) | 31.59 | 23.57 | 37.32 | 17.51 | 40.75 | 45.01 | 50.41 | 25.37 |
| | ITI (Mass Mean Shift) | 29.46 | 23.73 | 26.35 | 18.66 | 38.95 | 43.04 | 25.20 | **19.15** |
| | CAD | 37.97 | 19.64 | 41.57 | 19.80 | 52.86 | 35.52 | 56.96 | 22.59 |
| | IRCAN | 39.12 | 18.99 | 45.01 | 24.88 | 42.72 | 37.64 | 49.26 | 30.11 |
| | IRCAN-CAD | **41.90** | **17.35** | **48.61** | 19.48 | 51.55 | 31.42 | **57.77** | 22.09 |
| ECARE_KRE | Original | 75.49 | 24.51 | 55.04 | 44.96 | 57.40 | 42.60 | 68.90 | 31.10 |
| | Based_on | 75.59 | 24.41 | 61.55 | 38.45 | 59.10 | 40.90 | 67.86 | 32.14 |
| | Based_on_Formatted | 76.72 | 23.28 | 63.15 | 36.85 | 69.09 | 30.91 | 68.61 | 31.39 |
| | Utilizing_Formatted | 76.44 | 23.56 | 60.98 | 39.02 | 68.99 | 31.01 | 66.16 | 33.84 |
| | Opin | 63.52 | 36.48 | 55.04 | 44.96 | **73.80** | **26.20** | 57.12 | 42.88 |
| | ITI (Probe Weight Direction) | 73.04 | 26.96 | 49.58 | 50.42 | 60.51 | 39.49 | 73.42 | 26.58 |
| | ITI (Mass Mean Shift) | 73.80 | 26.20 | 47.60 | 52.40 | 49.58 | 50.42 | 71.44 | 28.56 |
| | CAD | 77.76 | 22.24 | 73.70 | 23.30 | 69.56 | 30.44 | 78.13 | 21.87 |
| | IRCAN | 77.38 | 22.62 | 76.06 | 23.94 | 57.87 | 42.13 | 69.84 | 30.16 |
| | IRCAN-CAD | **82.38** | **17.62** | **80.96** | **19.04** | 69.37 | 30.63 | **78.42** | **21.58** |

Table 2: Results (in %) of the multiple-choice task on the COSE_KRE and ECARE_KRE datasets.

this method identifies a direction in the activation space associated with factually correct statements and shifts activations along this direction during inference, thereby enhancing the truthfulness of LLMs. Analogous to its experimental setup, for each sample in each dataset, we concatenated the question/answer together and extracted head activations at the last token to collect a probing dataset. We then used the ITI method to identify the direction and intervened activations along this direction. We implemented two intervention directions in our experiments, i.e., Probe Weight Direction and Mass Mean Shift. **Context-aware decoding (CAD)** [36]**:** it adjusts the output probabilities of LLMs to emphasize differences when the context is utilized versus when it is absent. Specifically, this approach reduces the weighting of prior knowledge in favor of more pertinent contextual information. Since this method and our approach work in completely different ways: CAD manipulates the model's final output probabilities, while IRCAN focuses on identifying and enhancing neurons responsible for processing the context, these two methods can be seamlessly combined without any obstacles. In our experiments, we also report the performance of the combined system, which is referred to as IRCAN-CAD in our experiments. **Opin** [49]**:** this method argues that transforming factual questions into questions seeking opinions allows people to pay more attention to the context. Therefore, it uses opinion-based prompts to enable the model to generate context-faithful responses. Since the prompts designed by Opin do not apply to the completion task, we used this method as a baseline only on the multiple-choice datasets. To ensure a fair comparison, we adopted the same method for selecting the generated answer as described in §4.2 for Opin.

## 4.4 Main Results

We treat the number of identified context-aware neurons $h$ (§3.2), ranging from 1 to 16, and the enhancement strength $\beta$ for these neurons (§3.3), ranging from 2 to 20, as hyperparameters. All

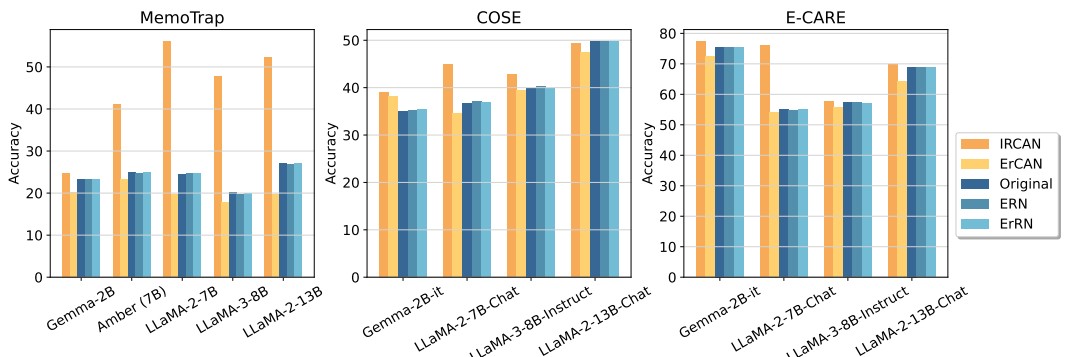

Figure 2: The results of ablation studies to illustrate the accuracy implications of different interventions. **ErCAN** denotes the variant where context-aware neurons are erased. **ERN** represents the enhancement of random neurons. **ErRN** indicates the erasure of random neurons.

datasets were randomly divided into validation and test sets in a 1:1 ratio. We employed grid search to identify the optimal hyperparameter configuration that maximized performance on the validation set. Subsequently, we report the test set results obtained using the identified optimal hyperparameter combination in Table 1 and Table 2.

**Completion Task**    The main results on the MemoTrap dataset are shown in Table 1. IRCAN significantly outperforms all the baselines. Intervening along the Mass Mean Shift direction brings different degrees of performance degradation to the vast majority of LLMs. Even more, the interventions cause Gemma-2B and Amber to completely fail to respond normally. Improvements along the Probe Weight Direction are also limited. This suggests that finding a direction relevant to the context and shifting activations along this direction during inference might not be a good way to enhance the attention of an LLM to contextual knowledge. The CAD method shows a slightly better improvement, but there is still a significant gap compared to our IRCAN. Notably, IRCAN achieves remarkable improvements of 129% for LLaMA-2-7B and 136% for LLaMA-3-8B in terms of accuracy when compared to the Original baseline. Such substantial performance improvements, achieved through strengthening merely a few or a dozen neurons, indicate that the neurons identified by our method play a pivotal role in processing context.

Additionally, the significant decrease in the SR metric observed for IRCAN suggests that the augmentation of context-aware neuron weights facilitates the utilization of knowledge derived from the provided context for the LLMs. Concurrently, this adjustment allows the model to substantially disregard the intrinsic knowledge embedded within its parameters. This indicates a shift in the model's reliance from pre-stored information to dynamically acquired context, improving its adaptability and accuracy in real-time processing.

Furthermore, when our method is integrated with the CAD approach, it leads to additional performance improvements over CAD across all models tested. This substantiates the complementary characteristics of our proposed methodology and the CAD approach. It implies that these two distinct strategies can collectively amplify the models' capacity to harness the information embedded within the context in unique and beneficial ways.

It is also notable that, in the Original setting, LLaMA-3-8B achieves the lowest ACC, along with the highest SR. This observation may seem somewhat counterintuitive. We believe it is due to the fact that LLaMA-3-8B, trained on extensive, high-quality multi-source data, has acquired more extensive world knowledge and tends to rely more on its pre-stored intrinsic knowledge when generating responses.

**Multiple-choice Task**    As presented in Table 2, ITI still performs poorly on the multiple-choice task. Moreover, only instructing LLMs to pay more attention to the knowledge in the context is not sufficient to enhance the model's utilization of contextual knowledge. Furthermore, the prompts with the best performance differ for different datasets and different models. Therefore, the requirement of meticulous prompt engineering undermines their generalizability.

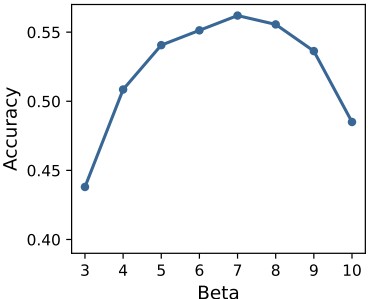

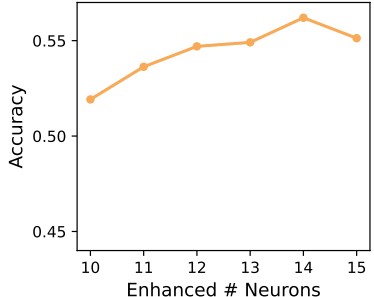

Figure 3: Model performance with different enhancement strengths $\beta$.

Figure 4: Model performance with different enhanced # neurons $h$.

In contrast, IRCAN significantly enhances the performance of LLMs in resolving knowledge conflicts in the multiple-choice task. This improvement is observed across the COSE_KRE and ECARE_KRE datasets for the majority of the evaluated LLMs. The multiple-choice task poses a greater challenge compared to the MemoTrap dataset, where the context directly provides the required knowledge for generation. In the multiple-choice task, LLMs are required to interpret and reason based on the implicit knowledge in the context to facilitate generation. Consequently, the improvements in performance observed for the two baseline models and our proposed IRCAN are less pronounced on this dataset than those noted on MemoTrap. Nonetheless, IRCAN still achieves a gain in accuracy and a drop in SR for the majority of LLMs engaged in this task, setting new state-of-the-art results.

We also observed that IRCAN does not perform as effectively for more capable models, such as LLaMA-3-8B-Instruct and LLaMA-2-13B-Chat. This may be attributed to the fact that the context in the examples of the COSE_KRE and ECARE_KRE datasets is generated by ChatGPT, which could potentially contain inaccurate information. Consequently, IRCAN struggles to precisely identify context-processing neurons in these datasets. Enhancing these neurons fails to help LLMs accurately follow context, instead relying more on their internal knowledge.

## 4.5  Ablation Studies

To investigate the importance of context-aware neurons, we conducted a series of ablation experiments. Initially, we examined the impact on model accuracy by erasing the detected context-aware neurons. Specifically, we set the weights of these neurons to 0 to deactivate them during the forward pass. We also performed a comparative analysis by randomly enhancing or erasing the same number of neurons as implemented in IRCAN. To ensure the reliability and robustness of our experimental results, we replicated the experiments 10 times independently and reported the average of these results as the foundation for our final analysis. This was designed to minimize selection bias and to reinforce the statistical significance of our findings.

The outcomes of these ablation studies illustrated in Figure 2 indicate a substantial drop in accuracy when context-aware neurons are deactivated, compared to the results of IRCAN. However, no matter whether erasing or enhancing random neurons, the performance remains similar to that of the original model. This suggests the critical role of our detected context-aware neurons in resolving knowledge conflicts, thereby validating their importance in the functionality of the model.

## 5  Analysis

### 5.1  Effect of Enhancement Strength

The relationship between the accuracy of the LLaMA-2-7B model on the MemoTrap dataset and the enhancement strength $\beta$ is depicted in Figure 3, where the number of enhanced neurons is fixed at 14. For further results on various enhancement strengths $\beta$, please refer to Figure 11 in Appendix D. It can be observed that as the enhancement strength for the context-aware neurons increases, model performance gradually improves, highlighting the pivotal role of the neurons identified by our method. Then, consistent with our intuition, performance begins to decline beyond a certain enhancement strength (7 in this scenario). This decline could be due to excessively high enhancements of certain neurons, leading to uncontrollable outputs or a reduction in model capabilities.

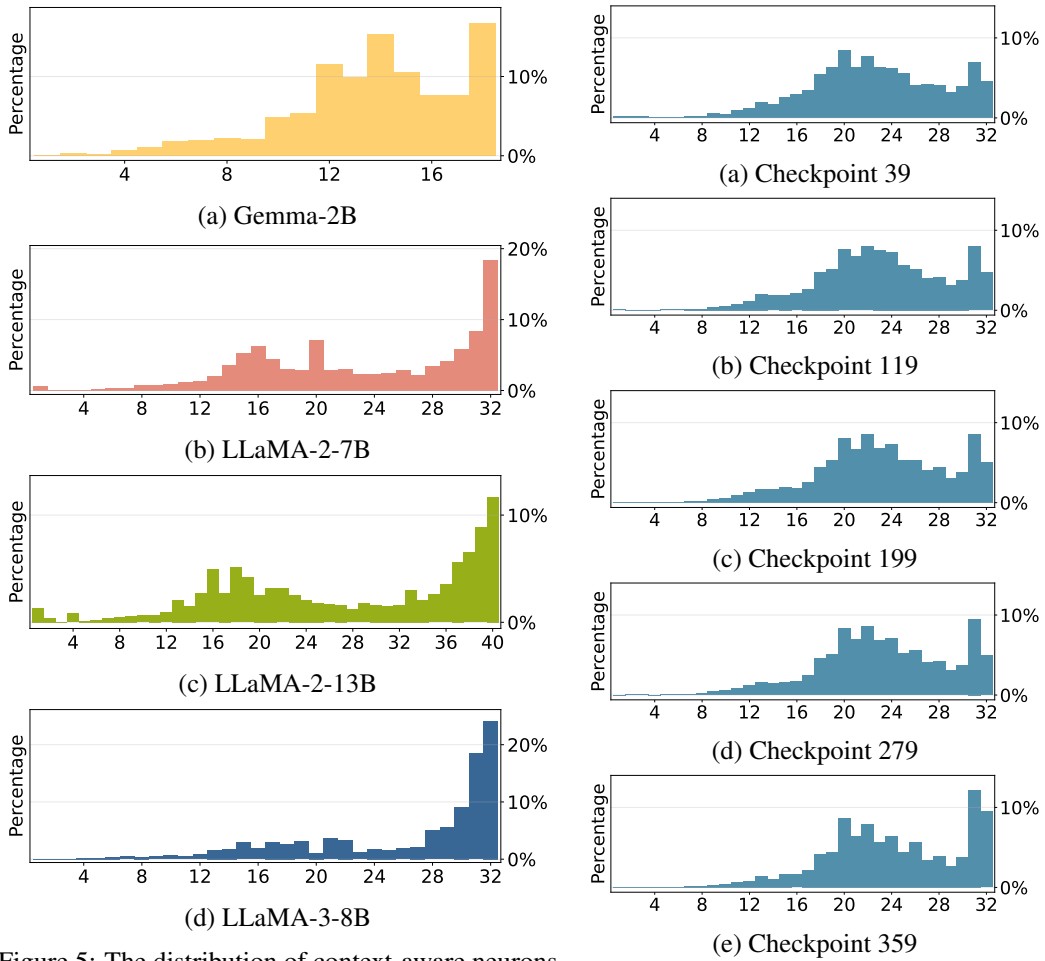

Figure 5: The distribution of context-aware neurons across layers with various LLMs.

(a) Gemma-2B

(b) LLaMA-2-7B

(c) LLaMA-2-13B

(d) LLaMA-3-8B

(a) Checkpoint 39

(b) Checkpoint 119

(c) Checkpoint 199

(d) Checkpoint 279

(e) Checkpoint 359

Figure 6: The distribution of context-aware neurons across layers with Amber's 5 checkpoints.

## 5.2 Effect of the Number of Context-Aware Neurons

The impact of the number of enhanced neurons $h$ on the performance of LLaMA-2-7B on the MemoTrap dataset is shown in Figure 4. We present results where the enhancement multiplier is fixed at 7, with comprehensive results available in Appendix D (see Figure 12). Similarly, observations reveal that as the number of enhanced neurons increases, the model's accuracy initially improves but subsequently begins to decline, resonating with the results observed with the enhancement strength.

## 5.3 Layer Distribution of Context-Aware Neurons

We illustrate the distribution of the candidate set (§3.2) of context-aware neurons identified by IRCAN across layers of Gemma-2B, LLaMA-2-7B, LLaMA-2-13B, and LLaMA-3-8B in Figure 5. Additionally, Figure 6 depicts the changes in the distribution of these identified context-aware neurons across layers of Amber during different training stages.[3] Overall, the context-aware neurons are primarily located in the top layers, with a relatively small portion in the intermediate layers. This aligns with prior findings that language models predominantly encode "semantic" information in the top layers [38, 5]. Notably, when comparing the distributions of LLaMA-2-7B and LLaMA-2-13B with LLaMA-3-8B, we observe that the context-aware neurons of LLaMA-3-8B exhibit a more prominent aggregation in the top layers, as LLaMA-3-8B was trained on significantly more data than

---

[3]Amber offers 360 checkpoints (0 to 359) from various training stages. Checkpoint 359 is the final one. Five evenly distributed checkpoints were selected for experiments.

| Models | | ARC | HellaSwag | MMLU | TruthfulQA | Winogrande | GSM8K | Average |
|---|---|---|---|---|---|---|---|---|
| Gemma-2B | Original | 48.29 | 71.13 | 40.99 | 33.02 | 66.38 | 17.66 | 46.25 |
| | IRCAN | 48.29 | 71.42 | 39.91 | 33.58 | 65.11 | 18.12 | 46.07 |
| Amber | Original | 43.09 | 73.34 | 23.99 | 33.98 | 66.38 | 3.49 | 40.71 |
| | IRCAN | 42.24 | 73.42 | 24.61 | 34.22 | 66.46 | 3.03 | 40.66 |
| LLaMA-2-7B | Original | 51.96 | 78.18 | 45.95 | 38.97 | 74.19 | 13.57 | 50.47 |
| | IRCAN | 52.56 | 77.15 | 46.35 | 37.89 | 73.01 | 12.66 | 49.94 |
| LLaMA-2-13B | Original | 57.59 | 81.72 | 54.94 | 36.90 | 76.01 | 23.12 | 55.05 |
| | IRCAN | 55.46 | 78.74 | 55.40 | 38.25 | 76.87 | 12.36 | 52.85 |
| LLaMA-3-8B | Original | 57.76 | 81.10 | 65.14 | 43.88 | 77.51 | 50.72 | 62.69 |
| | IRCAN | 56.48 | 80.86 | 64.56 | 45.08 | 75.61 | 36.92 | 59.92 |
| Gemma-2B-it | Original | 44.54 | 61.74 | 36.97 | 45.85 | 61.64 | 4.85 | 42.60 |
| | IRCAN | 44.54 | 61.79 | 37.38 | 45.86 | 61.33 | 5.00 | 42.65 |
| LLaMA-2-7B-Chat | Original | 51.79 | 77.73 | 47.39 | 45.32 | 72.53 | 22.97 | 52.96 |
| | IRCAN | 51.79 | 77.78 | 45.74 | 45.45 | 72.61 | 22.21 | 52.60 |
| LLaMA-3-8B-Instruct | Original | 61.34 | 78.04 | 65.83 | 51.69 | 75.69 | 75.36 | 67.99 |
| | IRCAN | 60.84 | 77.98 | 57.79 | 52.18 | 76.01 | 74.00 | 66.47 |
| LLaMA-2-13B-Chat | Original | 58.53 | 81.56 | 53.57 | 43.96 | 74.35 | 34.65 | 57.77 |
| | IRCAN | 58.62 | 81.58 | 53.63 | 43.94 | 74.43 | 34.80 | 57.83 |

Table 3: Results of general abilities of LLMs on widely-used benchmarks.

LLaMA-2-7B and LLaMA-2-13B. This observation is further substantiated by the distribution of context-aware neurons across five distinct checkpoints of Amber.

We also analyzed the overlap of identified context-aware neurons across different prompts. Results (Appendix E) show a high degree of overlap, demonstrating IRCAN consistently identifies these neurons regardless of the prompt.

## 5.4 Evaluation of General Abilities of LLMs

To investigate whether up-weighting context-aware neurons impairs the model's general abilities, we conducted evaluations of IRCAN on six widely-used benchmarks. These benchmarks are used in the widely-recognized Open LLM Leaderboard,[4] including ARC [6], HellaSwag [47], MMLU [15], Winogrande [34], GSM8K [7], and TruthfulQA [24]. We describe the experimental details in Appendix F. The experimental results, as shown in Table 3, reveal that IRCAN rarely impacts the general ability of the LLMs. Surprisingly, in some cases, it even leads to a slight performance improvement. These results suggest that IRCAN can reliably improve the capability of the LLMs in addressing knowledge conflict tasks while maintaining their excellent general capabilities.

## 6 Conclusion and Future Work

In this paper, we have presented IRCAN, a framework designed to mitigate knowledge conflicts in LLMs by identifying and reweighting context-aware neurons. Our extensive experiments across various models and tasks demonstrate that IRCAN significantly improves the fidelity of models to contextual knowledge. By enhancing context-aware neurons, IRCAN not only boosts model performance but also integrates seamlessly with existing methods, achieving state-of-the-art results in both completion and multiple-choice tasks. This work marks a significant step towards more reliable and nuanced AI systems capable of accurate context-sensitive information processing.

**Limitations** Our current study has only experimented on a few synthetic datasets. However, exploring the effectiveness of IRCAN in scenarios such as long-context tasks and RAG is also interesting and valuable. For instance, by enhancing the model's sensitivity and fidelity to retrieved documents in context, IRCAN is expected to improve the performance of generation models in RAG systems, enabling more accurate and contextually relevant text generation. We leave this for our future work.

---

[4]https://huggingface.co/spaces/HuggingFaceH4/open_llm_leaderboard

## Acknowledgements

The present research was partially supported by the National Key Research and Development Program of China (Grant No. 2023YFE0116400). We would like to thank the anonymous reviewers for their insightful comments.

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

# A Datasets

Examples of the datasets used in our experiments are shown in Table 4.

| | MemoTrap |
|---|---|
| $c$ | Write a quote that ends in the word "returned": |
| $q$ | Long absent, soon |
| gold answer | returned |
| | **COSE_KRE** |
| $c$ | Doctors' offices often provide magazines and other printed materials for patients to read while waiting for their appointments. |
| $q$ | Where would you find magazines along side many other printed works? |
| choices | [doctor, bookstore, market, train station, mortuary] |
| gold answer | A |
| | **ECARE_KRE** |
| $c$ | The passage of time can lead to significant changes in societal conditions, such as financial crises, which can subsequently impact mental health and suicide rates. |
| $q$ | After the financial crisis, the suicide rate increased significantly. What is the more possible cause of this? |
| choices | [The financial crisis left many people homeless., Time goes on.] |
| gold answer | B |

Table 4: An illustration of the example in each dataset.

# B Prompts for LLMs

The input prompt to LLMs for this task consists of the necessary instruction (e.g., "Choose the correct option to answer the following question."), context, question, and guiding suffix (e.g., "The correct answer is"). Through observation of the responses from various LLMs and multiple trials, we customized different suffixes aligned with their respective generative styles of each model, aiming to prompt them to immediately output the correct option in the continuation of the prompts. The prompts employed in our experiments across various models are illustrated in Figures 7 to 10.

---

**LLaMA-2-7B-Chat**

Choose the correct option to answer the following question:
{context}
{question}
A. {choice_A} B. {choice_B} C. {choice_C} ······
Answer:

---

Figure 7: Prompts used for LLaMA-2-7B-Chat.

---

**LLaMA-2-13B-Chat**

Choose the correct option to answer the following question:
{context}
{question}
A. {choice_A} B. {choice_B} C. {choice_C} ······
Answer:

---

Figure 8: Prompts used for LLaMA-2-13B-Chat.

| Gemma-2B-it |
|---|
| Choose the correct option to answer the following question:
{context}
{question}
A. {choice_A} B. {choice_B} C. {choice_C} ······
Answer:
The correct option is
** |

Figure 9: Prompts used for Gemma-2B-it.

| Llama-3-8B-Instruct |
|---|
| Choose the correct option to answer the following question:
{context}
{question}
A. {choice_A} B. {choice_B} C. {choice_C} ······
The correct answer is |

Figure 10: Prompts used for LLaMA-3-8B-Instruct.

## C   Details of the Prompt Engineering Methods

The original prompt and three deliberately designed prompts that instruct LLMs to pay more attention to the knowledge in the context are shown in Table 5. Modifications relative to the original prompt are highlighted in bold.

| Original Prompt |
|---|
| Choose the correct option to answer the following question:
{context}
{question}
A. {choice_A} B. {choice_B} C. {choice_C}... ...
... ... |
| Based_on Prompt |
| Choose the correct option to answer the following question **based on the context**:
{context}
{question}
A. {choice_A} B. {choice_B} C. {choice_C}... ...
... ... |
| Based_on_Formatted Prompt |
| Choose the correct option to answer the following question **based on the context**:
**Context:** {context}
**Question:** {question}
**Choices:** A. {choice_A} B. {choice_B} C. {choice_C}... ...
... ... |
| Utilizing_Formatted Prompt |
| Choose the correct option to answer the following question **utilizing the knowledge in the context**:
**Context:** {context}
**Question:** {question}
**Choices:** A. {choice_A} B. {choice_B} C. {choice_C}... ...
... ... |

Table 5: Prompts used in the prompt engineering based methods.

# D Effect of the Hyperparameters

We conducted experiments to explore the effect of hyperparameters on model performance. Figure 11 shows variations due to changes in enhancement strength $\beta$ and Figure 12 details changes associated with the number of enhanced neurons $h$. We intercept the results for $\beta$ from 3 to 10 and $h$ from 10 to 15 to show the results.

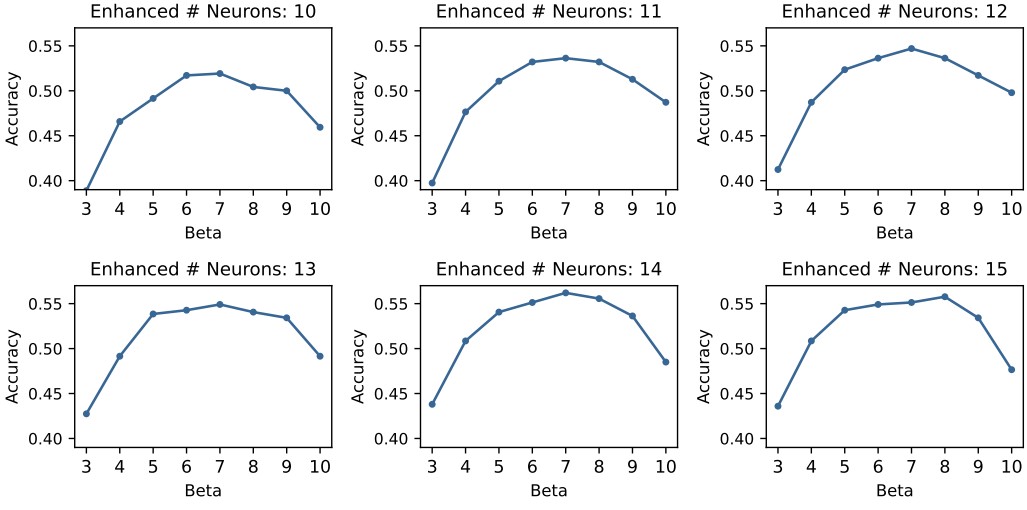

Figure 11: Model performance with different enhancement strengths $\beta$.

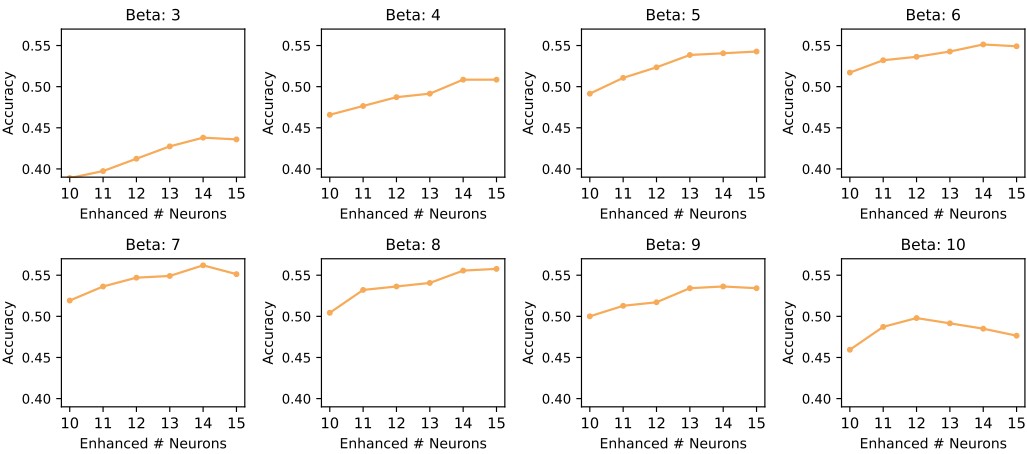

Figure 12: Model performance with different numbers of enhanced neurons $h$.

# E Context-Aware Neuron Intersection for Different Prompts

We conducted experiments to validate the robustness of identified context-aware neurons to different prompts. Specifically, we conducted experiments on the COSE dataset to identify context-aware neurons using prompts different from those used in IRCAN for each model. We displayed the intersection of the top 300 neurons identified across different prompts, as shown in the Table 6. The results, illustrated in Figure 13, reveal that over 50% of the neurons identified by IRCAN coincide with those detected using alternative prompts. This significant overlap substantiates the efficacy and robustness of IRCAN in consistently identifying neurons that process contextual information across diverse prompts.

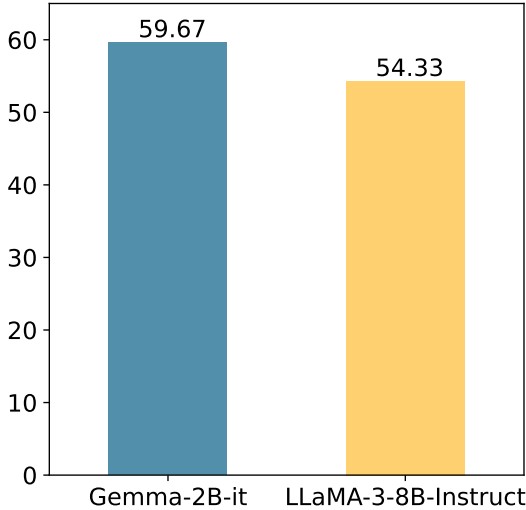

Figure 13: The intersection of neurons identified with different prompts was used for Gemma-2B-it and LlaMA-3-8B-Instruct.

## F    Experimental Details of General Ability Evaluation

We use the Eleuther AI LM Evaluation Harness[5] to conduct our general ability evaluation experiments. For the ARC, HellaSwag, MMLU, Winogrande, and GSM8K benchmarks, a 5-shot setting was employed, while a zero-shot setting was utilized for the TruthfulQA assessment. Metrics for evaluation varied with each benchmark: acc_norm for ARC and HellaSwag; acc for Winogrande, MMLU, and TruthfulQA (truthfulqa-mc2); and strict exact_match for GSM8K, which is consistent with the Open LLM Leaderboard. Moreover, an average of the results across these six benchmarks was calculated.

## G    Computational Cost

Our experiments for identifying context-aware neurons can be run on a single A100 GPU with 80 GB of memory. The duration required for these experiments depends on the scale of model parameters, the size of the dataset, and the length of individual examples within the dataset. Overall, the time consumption is entirely acceptable. Taking the COSE_KRE dataset as an example, the neuron identification experiment on the smallest model (2B parameters) completes in less than 2 hours, and the experiment on the largest model (13B parameters) takes less than 18 hours.

Importantly, the IRCAN does not introduce any additional inference time costs. The identification and reweighting of context-aware neurons are performed offline, allowing the modified model to be directly utilized during online testing. As a result, the inference process remains unaffected in terms of computational time.

---

[5]https://github.com/EleutherAI/lm-evaluation-harness

| | Gemma-2B-it |
|---|---|
| Prompt 1 | Choose the correct option to answer the following question: 
 {context} 
 {question} 
 A. {choice_A} B. {choice_B} ... ... 
 Answer: 
 The correct option is 
 ** |
| Prompt 2 | Choose the correct option to answer the following question: 
 {context} 
 {question} 
 A. {choice_A} B. {choice_B} ... ... 
 Answer: |
| | Llama-3-8B-Instruct |
| Prompt 1 | Choose the correct option to answer the following question: 
 {context} 
 {question} 
 A. {choice_A} B. {choice_B} ... ... 
 The correct answer is |
| Prompt 2 | Choose the correct option to answer the following question: 
 {context} 
 {question} 
 A. {choice_A} B. {choice_B} ... ... 
 Answer: |

Table 6: Prompts used in the context-aware neuron intersection experiment.

