# OpenReview forum: "IRCAN: Mitigating Knowledge Conflicts in LLM Generation via Identifying and Reweighting Context-Aware Neurons"
_NeurIPS.cc/2024/Conference — NeurIPS 2024 poster_

### Official Review · Reviewer_CBV2 · 2024-07-10

**Soundness:** 3
**Presentation:** 4
**Contribution:** 3
**Rating:** 8
**Confidence:** 4

**Summary:**

This work presented a framework IRCAN to locate key neurons for processing contextual cues, thereby mitigating conflicts between knowledge obtained from pre-training and knowledge within the context. Experiments on completion and multi-choice tasks showed that the IRCAN benefits the base model in knowledge conflict tasks.

**Strengths:**

1. This paper is well-written and easy to read, and the authors present their methods and experiments very clearly.
2. The paper innovatively addresses the knowledge conflict issues by manipulating neurons. Experimental results validate the effectiveness of the method. Particularly, the improvement in results on knowledge completion tasks is quite significant.
3. Ablation analysis is sufficient for understanding the proposed method in depth.

**Weaknesses:**

1. While the paper has demonstrated the effectiveness of the proposed IRCAN on 3 datasets, its performance on additional datasets remains unexplored. It would be beneficial for the authors to include more knowledge-related datasets in future versions of the paper to further validate the generalizability of the model.
2. In the completion task, the experiments compared the accuracy of the proposed IRCAN and baselines, and the IRCAN improved the performance by a large margin. However, the experiments are conducted on merely one dataset, weakening the generalization of the IRCAN, and there is no computing comparison, for example, GPU time-consuming.

**Questions:**

1. This paper assumed that contextual knowledge was more reliable than parametric knowledge, and what if the context itself was fake or misleading?
2. How did the authors construct knowledge datasets along with the context?
3. The paper mentioned: “To calculate the attribution score Attr(n^{l}_{i}), we gradually change the activation value of a neuron n^{l}_{i} from v^{l}_{q_{i}} to v^{l}_{(c,q)_{i}}…”, what are the details of the activation value’s change process?
4. In section 3.3, what is the object W(n^{l}_{i}) for the reweighting? Activation value or the attention matrix?
5. For the evaluation of general abilities, results in Table 3 show that IRCAN can cause other knowledge-related tasks like MMLU and Winograd to fluctuate. What could be the cause of this fluctuation? Does that demonstrate that emphasizing the context may bring knowledge degradation?

**Limitations:**

Various LLMs were employed in this paper, however, most of the models were 7B/8B. The authors could validate the proposed IRCAN in larger LLMs to complete the work in the future.

---

> ### Author Rebuttal · Authors · 2024-08-07
>
> We are deeply grateful for your positive assessment of our work and the recognition of the value in our work. Your feedback is highly encouraging and valuable to us. We will address each of your concerns and questions in detail:
>
> **Re to W1:** **IRCAN’s performance on additional knowledge-related datasets remains unexplored.**
>
> Thank you for your insightful comments. We highly agree with your suggestion that incorporating more knowledge-related datasets would be beneficial for further validating the generalizability of our model. Indeed, in the ongoing process of our research, we have been actively investigating the existence of such datasets that include new correct knowledge in context, while the corresponding knowledge encoded in the LLM is outdated or incorrect. For example, an LLM trained by December 19, 2022 (the date of the Qatar World Cup final) would only have knowledge that Argentina has won the World Cup twice. If we present the results of the 2022 World Cup in the context and construct question-answering tasks, such a dataset would provide an excellent means to further validate the effectiveness and generalization of our model. However, to the best of our knowledge, such datasets are currently unavailable, which hinders the further validation of our IRCAN. We look forward to the emergence of such datasets and would like to collect more suitable datasets for our framework in the future.
>
> **Re to W2 (1): The experiments are conducted on merely one completion dataset.**
>
> Thank you for your valuable feedback. To the best of our knowledge, other datasets involving knowledge conflicts for completion tasks are not available. We are willing to validate IRCAN on more completion datasets if they are available in the future.
>
> **Re to W2 (2): There is no computing comparison.**
>
> We are grateful for your suggestion. Although we have stated and discussed the computational resources and time consumption required for our method in Appendix F, we will also include a time-consumption comparison with other methods in the next version of the paper as Section 5.1.
>
> **Re to Q1:** **This paper assumed that contextual knowledge was more reliable than parametric knowledge, and what if the context itself was fake or misleading?**
>
> Thank you for your insightful comments. The issue you mentioned is precisely what we have considered during the course of this work. In our future work, we plan to delve deeper into this research. Specifically, we intend to propose a framework that first incorporates a judgment mechanism to determine whether to focus more on contextual knowledge or adhere to the internal knowledge of the LLM, and then enhances the fidelity to the chosen aspect during generation.
>
> **Re to Q2: How did the authors construct knowledge datasets along with the context?**
>
> As described in Section 4.1 of our paper, we used publicly available datasets, MemoTrap, COSE_KRE and ECARE_KRE, in our experiments.
>
> The MemoTrap [1] dataset is created by replacing the ending words of common proverbs with other words, and then prompting the model with the context instruction: "Write a quote that ends with the word '{the replaced word}'". It evaluates the models’ ability to adhere to the given context to complete an unfamiliar phrase, rather than defaulting to a well-known phrase that has been encoded in its parameters during training.
>
> The COSE_KRE and ECARE_KRE [2] datasets are respectively derived from the ECQA and e-CARE datasets. The derivation process involves selecting one of the incorrect answer choices and prompting ChatGPT to generate explanations supporting this incorrect answer. Specifically, the selected incorrect answer is treated as the correct answer, and the generated explanation is used as the context for the multiple-choice question.
>
> [1] Liu et al. The MemoTrap Dataset, 2023. https://github.com/inverse-scaling/prize/tree/main/data-release.
>
> [2] Ying et al. Intuitive or Dependent? Investigating LLMs' Behavior Style to Conflicting Prompts. CoRR, abs/2309.17415, 2023.
>
> **Re to Q3:** **What are the details of the activation value's change process?**
>
> For simplicity, let us denote the difference of {v}_{(c,q)}_{i}^{l} minus {v}_{q}_{i}^{l} as z. We divide z into m = 20 equal parts. The activation value’s change process involves performing 20 forward propagations, each time replacing the model activation with
>
> ${\boldsymbol{v}_{q}}_i^l + \frac{1}{20}z$,
>
> ${\boldsymbol{v}_{q}}_i^l + \frac{2}{20}z$,
>
>  …
>
> ${\boldsymbol{v}_{q}}_i^l + z$
>
> respectively. Finally, we compute the cumulative sum according to Equation (3) in our paper to obtain the attribution score.
>
> However, in practical implementation, we repeat each example 20 times to form a batch, and also use the 20 activation values mentioned above as a batch to replace to the model. This allows us to complete the process with a single forward pass.
>
> **Re to Q4: In section 3.3, what is the object W(n^{l}_{i}) for the reweighting? Activation value or the attention matrix?**
>
> The object we reweight is the weights of the MLP layer.
>
> **Re to Q5:** **Cause of the fluctuation in experimental results in Table 3.**
>
> As we have enhanced certain neurons, it is expected and normal for the model's outputs to change. As demonstrated in Table 3 in our paper, the fluctuation amplitude of the results is very small, which is entirely acceptable. Even minor modifications to the model's decoding hyperparameters or the use of different decoding methods can cause significant variations in the generated results. However, these variations do not indicate an increase or decrease in the model's inherent capabilities.
>
> **Re to Limitations: Validation on larger LLMs.**
>
> Thank you for your great suggestion. We must emphasize that the experiments in our paper have been conducted on 13B models for both the completion task and the multiple choice task, in addition to the 7B/8B scale model. In the future, we will validate the proposed IRCAN on LLMs with larger scales, e.g., 70B.

---

> > ### Comment · Reviewer_CBV2 · 2024-08-14
> >
> > Thank you for the detailed responses, my concerns were addressed, and I will change my previous score.

---

> > > ### Author Response · Authors · 2024-08-14
> > >
> > > We sincerely thank you for your constructive feedback and for the revised score. Your insightful comments and suggestions have been instrumental in refining the quality of our paper.

---

### Official Review · Reviewer_NbaM · 2024-07-11

**Soundness:** 3
**Presentation:** 3
**Contribution:** 3
**Rating:** 6
**Confidence:** 4

**Summary:**

The paper addresses the valuable problem of mitigating parametric and contextual knowledge conflicts in LLM generation with a novel and reasonable method. It is well-written, with a comprehensive experimental design showing significant improvements in completion and multi-choice tasks. However, the evaluation is limited to short contexts, raising concerns about scalability to longer contexts. The assumption that contexts are contradictory to parametric knowledge may not always hold, and the method's performance on RAG tasks and its impact on inference speed needs further exploration. Additionally, unexpected performance results between llama3-8b and llama2-7b require explanation.

**Strengths:**

- S1: The studied problem of mitigating parametric and contextual knowledge conflicts is of great value
- S2: The paper is well-written and easy to follow
- S3: Though simple, the proposed method is novel and reasonable for mitigating knowledge conflicts
- S4: The experimental design is comprehensive and rigorous, and the results show significant improvement in completion and multi-choice tasks
- S5: The discussion and results on preserving performances on other tasks are highly appreciated

**Weaknesses:**

- W1: The evaluation is limited to datasets with relatively short contexts, it is questionable whether the proposed method can scale to long contexts.
- W2: The paper assumes that the contexts are contradictory to the parametric knowledge. However, in many cases, only a tiny fraction of the context is inconsistent with the parametric knowledge, while the others are consistent or unknown. Is the proposed method still valid under these situations?
- W3: The (partly) contradiction may frequently occur in RAG, it would be great to know how this method performs on RAG tasks.
- W4: It seems the proposed method has to select neurons first before answering and this involves forward passes once with and once without contexts, will this process make the inference significantly slower? Some results and discussions on time complexity would be highly appreciated.
- W5: As shown in Table 1, llama3-8b performs worse than llama2-7b, which is not as expected. Is there any explanation for this?
- W6: Are the salient neurons selected specifically for each question (with or without context), or are shared across data points like this: https://arxiv.org/abs/2311.15983, why or why not?

**Questions:**

Please refer to W1-W6.

**Limitations:**

Yes.

---

> ### Author Rebuttal · Authors · 2024-08-07
>
> Thank you so much for your valuable feedback and insightful comments. We will address each of your concerns and questions in detail:
>
> **Re to W1 & W3: Evaluations on datasets with long contexts and RAG tasks.**
>
> Thank you for this valuable feedback. We understand the prevalence of long-context inputs in real-world applications. In future work, we will continue to look for knowledge conflict datasets with long context and explore the effectiveness of our proposed method on such datasets.
>
> As discussed in Section 6 of our paper, we also acknowledge that applying our method on RAG tasks is a promising and practical direction. Specifically, by enhancing the model’s sensitivity and fidelity to retrieved documents in context, IRCAN is expected to significantly improve the performance of generation models in RAG systems, enabling more accurate and contextually relevant text generation.
>
> In subsequent work, we will explore IRCAN’s effectiveness in application scenarios such as long-context tasks and RAG.
>
> **Re to W2: Is the proposed method still valid in situations where the contextual knowledge is inconsistent with the parametric knowledge?**
>
> Thank you for your insightful comment. We believe our proposed method remains valid when the knowledge in the context does not conflict with the internal knowledge of LLMs.
>
> Our IRCAN enhances specific neurons crucial for processing contextual cues, ensuring LLMs generate outputs more faithful to the knowledge in the context. Moreover, the experiments in Section 5.4 demonstrate that IRCAN does not compromise other general capabilities. Therefore, when there is no conflict between the knowledge in the context and the knowledge inherent to the LLMs, enhancing these neurons solely improves the fidelity of the models to the contextual knowledge (i.e., internal knowledge), without other effects.
>
> **Re to W4 (1): Will the process of selecting neurons make the inference significantly slower?**
>
> Thank you for your valuable comments. The neuron selection process in our IRCAN does not lead to more inference time costs.
>
> **1.**  **First, we utilize some examples to identify context-aware neurons offline.** Specifically, for each example, we first calculate the attribution scores of neurons. Then, we select the top z neurons with the highest attribution scores as the candidate set for each example. Ultimately, we count the number of co-occurrences of neurons in all candidate sets, and we select the top h neurons with the highest number of co-occurrences as identified context-aware neurons. Therefore, identified context-aware neurons are shared across all data instances.
>
> **2.**  **Then, we reweight these context-aware neurons of the LLM.**
>
> **3.**  **During online testing, we take the modified model for inference, without adding any inference time cost.**
>
> We suspect that such misunderstanding may be caused by the lack of clarity of the expression in Section 3.2: "**Ultimately, we allow each example to vote for the candidate neurons based on their attribution scores, and we select the top h neurons that receive the most votes.**". We will revise this sentence to "**Ultimately, we count the number of co-occurrences of neurons in all candidate sets, and we select the top h neurons with the highest number of co-occurrences as identified context-aware neurons.**" in the next version of our paper. Moreover, at the end of Section 3.2, we will add "These context-aware neurons are shared across all data instances." to enhance the clarity of the paper.
>
> **Re to W4 (2): Some results and discussions on time complexity would be highly appreciated.**
>
> Due to page limitations, we have stated and discussed the computational resources and time consumption required for our method in Appendix F. In the revised version of our paper, we will move this content to the main body of the paper as Section 5.5.
>
> **Re to W5: Explanation for why llama3-8b performs worse than llama2-7b in Table 1.**
>
> Thank you for bringing this to our attention. As shown in Table 1, LLaMA-3-8B achieved lower accuracy (ACC) on the completion task, along with a higher stubbornness rate (SR). The SR measures how often the model's output matches common ending words of well-known proverbs. These results suggest that LLaMA-3-8B, trained on extensive, high-quality multi-source data, has acquired more extensive world knowledge and relies more on its pre-stored intrinsic knowledge when generating responses. We will incorporate this discussion into Section 4.4 in the next paper version.
>
> **Re to W6: Are the salient neurons selected specifically for each question (with or without context), or are shared across data points, why or why not?**
>
> Thank you for your insightful comments. We have thoroughly read the paper you mentioned, which employs a linear probing method to select salient neurons and then integrates them to serve as curated multi-layer features for text classification, effectively improving text classification accuracy, efficiency, and interpretability. The neurons we selected are the same as those in this work, which are shared across data points.
>
> Our rationale for adopting this neuron selection method is twofold:
>
> Firstly, we utilize some data to find the neurons responsible for processing the context in an offline setting, and then augment their weights in LLMs. During online inference, we can improve the model's attention to contextual knowledge in the input data without increasing inference time.
>
> Secondly, if for each example, the neurons are individually identified by the attribution scores computed through two forward propagations (once with and once without context), the identified neurons may be not all responsible for processing contexts, and may have poorer generalizability to other examples.

---

> > ### Comment · Reviewer_NbaM · 2024-08-13
> >
> > Thanks for the rebuttal and I will keep my evaluation.

---

> > > ### Author Response · Authors · 2024-08-14
> > >
> > > We sincerely appreciate the time and effort you have dedicated to reviewing our paper and our rebuttal. We are grateful for your constructive comments and valuable insights.

---

### Official Review · Reviewer_kYip · 2024-07-13

**Soundness:** 3
**Presentation:** 3
**Contribution:** 3
**Rating:** 6
**Confidence:** 3

**Summary:**

The paper proposed a new framework, IRCAN, to enable LLMs to pay more attention to new knowledge in context and generate context-sensitive outputs. The framework first identifies neurons that significantly contribute to context processing by utilizing a context-aware attribution score derived from integrated gradients and then reweighting these neurons. Experiments show the framework can effectively mitigate knowledge conflicts while not harming the general abilities of LLMs.

**Strengths:**

1. The proposed framework is novel and effective.
2. The analysis of the framework is comprehensive.
3. The paper is well written and easy to follow.

**Weaknesses:**

1. The authors state they have discussed the limitations in the checklist, but no limitation section is found.

**Questions:**

1. Why the paper does not compare the proposed framework with fine-tuning? How will instruction-tuning the model to be more sensitive to new knowledge in context help? While the proposed framework and fine-tuning both involve updating parameters and the proposed framework costs many hours that might not be much more efficient than fine-tuning, it seems natural to consider fine-tuning as a baseline.
2. What is the performance of instructing the LLM to pay more attention to the knowledge in context in the prompt? For the multi-choice task, it seems that simply instructing the model to adopt the knowledge in the context will be effective enough, which will cost much less compared to the framework.

**Limitations:**

The authors state they have discussed the limitations in the checklist, but no limitation section is found.

---

> ### Author Rebuttal · Authors · 2024-08-07
>
> Thank you very much for your insightful comments and valuable suggestions! We greatly appreciate you taking the time to review our work and provide constructive feedback to improve the quality of our paper. We will address each of your concerns and questions in detail:
>
> **Re to Weaknesses #1 & Limitations: No limitation section.**
>
> Thank you for this valuable feedback. We have discussed the limitation of our paper in Section 6, that is, the effectiveness of the proposed method on the retrieval-augmented generation (RAG) tasks was not verified, but it was not explicitly written as a limitation section. We are very sorry for the confusion. We will remove the second paragraph regarding RAG in Section 6 and add a limitation section as Section 7. We have drafted a preliminary version of the Limitation section below and would greatly appreciate your feedback:
>
> Our current study has only experimented on a few synthetic datasets, however, exploring the effectiveness of IRCAN in application scenarios such as long-context tasks and RAG is also necessary and valuable. For instance, by enhancing the model’s sensitivity and fidelity to retrieved documents in context, IRCAN is expected to significantly improve the performance of generation models in RAG systems, enabling more accurate and contextually relevant text generation. We will explore this in the future.
>
> **Re to Questions #1: Comparisons with instruction-tuning.**
>
> Thank you for bringing this to our attention. We fine-tune Gemma-2B and LLaMA-2-7B on the Conifer dataset proposed by Sun et al. [1] to improve the model's ability to follow complex instructions. To ensure data diversity, we mixed this dataset with the general SFT dataset ShareGPT [2] for training, following Sun et al. The original ShareGPT dataset comprises 93,336 examples, and after filtering out unavailable or low-quality data instances, the dataset size remains at 92,585. The data size of the Conifer dataset is 13606. We utilized 8 A100 GPUs to train each model for 3 epochs, with a maximum sequence length of 4,096 tokens during training. The training time is 3.25 hours for Gemma-2B and 11.75 hours for LLaMA-7B.
>
> Experiment results are reported in Table 4 in the attached PDF. During testing, we used two settings: one using the same prompt as our method (indicated as IT in Table 4) and the other adding the template used in training to the prompt (indicated as IT-template in Table 4). The experimental results show that the instruction-tuned LLMs struggle to handle the knowledge conflict task. Furthermore, compared to our method (which requires only one A100 GPU, taking 1.1 hours for Gemma-2B and 2.5 hours for LLaMA-7B), the instruction-tuning method requires a significantly larger training dataset and greater resource consumption, which is a substantial drawback.
>
> In addition, we must emphasize that, IRCAN, functioning as a post-processing neuron editing method, holds a unique advantage of **offering remediation for already trained models**.
>
> [1] Conifer: Improving Complex Constrained Instruction-Following Ability of Large Language Models. https://arxiv.org/abs/2404.02823.
>
> [2] The ShareGPT dataset. https://huggingface.co/datasets/anon8231489123/ShareGPT_Vicuna_unfiltered
>
> **Re to Questions #2:** **Comparisons with the experiment that instruct LLMs to pay more attention to the knowledge in context in the prompt.**
>
> We greatly appreciate your feedback. We curated three types of prompts to explicitly instruct LLMs to pay more attention to the knowledge in the context, and experimented on the multiple-choice task. Below are these prompts (the changes are highlighted in bold) and the original prompt used in our paper:
>
> **Original Prompt:**
>
> Choose the correct option to answer the following question:
>
> {context}
>
> {question}
>
> {choices}
>
> ……
>
> **Prompt 1:**
>
> Choose the correct option to answer the following question **based on the context**:
>
> {context}
>
> {question}
>
> {choices}
>
> ……
>
> **Prompt 2:**
>
> Choose the correct option to answer the following question **based on the context**:
>
> **Context:** {context}
>
> **Question:** {question}
>
> **Choices:** {choices}
>
> ……
>
> **Prompt 3:**
>
> Choose the correct option to answer the following question **utilizing the knowledge in the context**:
>
> **Context:** {context}
>
> **Question:** {question}
>
> **Choices:** {choices}
>
> ……
>
> Experimental results are shown in Table 3 in the attached PDF. We can observe that our method achieves the best performance overall. There remains a large gap between the performance achieved by these prompt engineering based methods and that obtained by IRCAN-CAD. This indicates that only instructing LLMs to pay more attention to the knowledge in the context is not sufficient to enhance the model's utilization of contextual knowledge. Moreover, the prompts with the best performance differs for different datasets and different models. Therefore, the requirement of meticulous prompt engineering damages their generalizability. We will add these results and further analyses to the next version of our paper.

---

> ### Comment · Reviewer_kYip · 2024-08-09
> **Thanks for your reply**
>
> Thanks for your reply. The new experiments address my concerns well. I have one more question about your limitation section.
> > For instance, by enhancing the model’s sensitivity and fidelity to retrieved documents in context, IRCAN is expected to significantly improve the performance of generation models in RAG systems
>
> In RAG systems, the retrieved text can sometimes be noisy. Would enhancing the model’s sensitivity and fidelity to retrieved documents help in these scenarios?

---

> ### Author Response · Authors · 2024-08-10
>
> Thank you for your prompt and insightful response. We appreciate the time and effort you've taken to review our rebuttal.
>
> The RAG technology supplements models by fetching external data in response to queries, thus ensuring more accurate and current outputs. In fact, in the field of RAG, researchers have recognized that noise in retrieved external data can adversely affect the quality of generated content. To address this, they often employ post-retrieval processing techniques to remove noise from the retrieved documents.
>
> For instance, some researchers incorporate a **re-ranking** stage subsequent to the initial retrieval process, where the retrieved documents are reassessed, scored, and reorganized to more effectively emphasize those most relevant to the query while diminishing the influence of less relevant ones. Methods such as sequence pair classification and re-scoring are introduced to re-rank documents [1-5], thereby improving the relevance between the retrieved content and the query.
>
> Additionally, some approaches involve **filtering** phase to remove documents that fail to meet specified quality or relevance standards. For example, FiD-TF [6] and RECOMP [7] focus on removing irrelevant or redundant tokens and information from retrieved documents. Self-RAG [8] introduces a self-reflection mechanism to efficiently filter out irrelevant content.
>
> Therefore, we believe that addressing the noise problem in retrieved text is a critical issue that the RAG field is actively tackling and one that the RAG community should indeed resolve. Our IRCAN method can complement these efforts by enhancing the model’s sensitivity and fidelity to re-ranked or filtered text, thereby further improving the effectiveness of RAG and its ability to deliver more accurate and current outputs.
>
>
>
> [1] Glass et al. Re2g: Retrieve, Rerank, Generate. NAACL 2022.
>
> [2] Dai et al. Promptagator: Few-shot Dense Retrieval From 8 Examples. ICLR 2023.
>
> [3] Ram et al. In-Context Retrieval-Augmented Language Models. TACL 2023.
>
> [4] Shao et al. Enhancing Retrieval-Augmented Large Language Models with Iterative Retrieval-Generation Synergy. EMNLP 2023.
>
> [5] Hofstätter et al. Fid-light: Efficient and effective retrieval-augmented text generation. SIGIR 2023.
>
> [6] Berchansky et al. Optimizing Retrieval-augmented Reader Models via Token Elimination. EMNLP 2023.
>
> [7] Xu et al. RECOMP: Improving Retrieval-Augmented LMs with Compression and Selective Augmentation. arXiv, abs/2310.04408.
>
> [8] Asai et al. Self-RAG: Learning to Retrieve, Generate, and Critique through Self-Reflection. arXiv, abs/2310.11511.

---

> ### Comment · Reviewer_kYip · 2024-08-11
>
> Thanks for your response. I will improve the score from 5 to 6. Please make sure these new experiments are appropriately included in the revised version. I would recommend the authors discuss the application scenarios of the proposed method in the limitation or discussion section.

---

> ### Author Response · Authors · 2024-08-11
>
> Thank you very much for your positive feedback and for improving the score of our paper. We sincerely appreciate your valuable comments and suggestions, which have greatly contributed to enhancing the quality of our work.

---

### Official Review · Reviewer_qHtz · 2024-07-14

**Soundness:** 3
**Presentation:** 3
**Contribution:** 2
**Rating:** 6
**Confidence:** 4

**Summary:**

The paper introduces a novel framework, IRCAN, aimed at addressing knowledge conflicts in Large Language Models (LLMs). By identifying and enhancing neurons that are crucial for processing contextual cues using an attribution score derived from integrated gradients, the framework significantly improves the generation of context-sensitive outputs. Tested across various models and tasks, IRCAN not only enhances model performance notably but also integrates seamlessly as a plug-and-play solution with existing models, establishing new performance benchmarks in handling knowledge conflicts.

**Strengths:**

1. The proposed method is plug-and-play and does not need additional training.
2. The authors conduct comprehensive experiments.
3. Steering the context-aware neurons seems effective on many large language models and many down-stream tasks.

**Weaknesses:**

1. Lack comparisons between other steering methods, like steering probed directions or steering SAE's features.
2. The propose methods will cost more inference time.
3. I think knowledge conflict also has its benefits. It may help LLMs defend some jailbreaking or harmful uses. So increase the model's faithfulness to its context may introduce some safety problems.

**Questions:**

Who you choose neurons? How do neurons compare with probed directions (https://arxiv.org/abs/2306.03341) or features from SAEs (https://transformer-circuits.pub/2024/scaling-monosemanticity/index.html#assessing-interp/) ?

**Limitations:**

As mentioned in weaknesses.

---

> ### Author Rebuttal · Authors · 2024-08-07
>
> We are deeply grateful for your positive assessment of our work and the recognition of the value in our work. Your feedback is highly encouraging and valuable to us. Below, we provide detailed answers to each of your concerns.
>
> **Re to Weaknesses #1 & Questions: Lack of comparisons with other steering methods, like steering probed directions or steering SAE's features.**
>
> Thank you for bringing this to our attention. The Inference-Time Intervention (ITI) method you mentioned identifies a direction in the activation space associated with factually correct statements and shifts activations along this direction during inference, thereby enhancing the truthfulness of LLMs.
>
> Analogous to their experimental setup, we conducted experiments on the completion and multiple-choice tasks to explore whether it is possible to find a direction related to perceiving and processing context, and whether it is possible to enhance LLMs' attention to contextual knowledge during generation by shifting activations along this direction. Similarly, for each sample in the MemoTrap or COSE_KRE dataset, we concatenate the question/answer, extract head activations at the last token to collect a probing dataset. Then we use the ITI to identify the direction and intervene activations. We implemented both intervention directions, i.e., Probe Weight Direction and Mass Mean Shift, and reported the results in Table 1 and Table 2 in the attached PDF.
>
> Experimental results show that intervening along the Mass Mean Shift direction significantly degrades the performance of most LLMs on both datasets, even more, causing models like Gemma-2B and LLaMA-3-8B to completely fail to respond normally. Improvements along the Probe Weight Direction are also limited, and our IRCAN still achieves the best performance.
>
> We will continue to complete experiments on the ECARE_KRE dataset. These experimental results and further analyses will be integrated into the revised version of our paper.
>
> As for the method of steering SAE's features as you suggested, we don't think it can be applied to our task. The SAE series of works aims to extract interpretable specific features from LLMs, such as features for famous people, features related to bias, etc. Furthermore, through feature steering, where they clamp specific features of interest to artificially high or low values during the forward pass, the output of the model can be **specifically** modified, and the **specific** behavior of the model can be controlled. For example, clamping the Transit infrastructure feature "1M/3" to 5× its maximum activation value causes the model to mention a bridge when it otherwise would not. Similarly, steering features related to bias can alter the model’s biases.
>
> However, in our work, the context may encompass a wide variety of knowledge (equivalent to a wide variety of features). Despite the varied nature of this contextual knowledge, our IRCAN approach successfully achieves the manipulation behavior of "enabling LLMs to generate outputs that are more faithful to the context". However, it would not be feasible to manipulate LLMs to produce outputs related to a broad spectrum of contextual knowledge by adjusting one or several SAE's features. Consequently, the SAE approach cannot be implemented on our task and is not suitable for comparison with our approach.
>
> **Re to Weaknesses #2: More inference time costs.**
>
> Thank you for your valuable feedback, but we have to emphasize that our IRCAN does not lead to more inference time costs.
>
> **1.**  **First, we utilize some examples to identify context-aware neurons offline.** Specifically, for each example, we first calculate the attribution scores of neurons. Then, we select the top z neurons with the highest attribution scores as the candidate set for each example. Ultimately, we count the number of co-occurrences of neurons in all candidate sets, and we select the top h neurons with the highest number of co-occurrences as identified context-aware neurons. Therefore, identified context-aware neurons are shared across all data instances.
>
> **2.**  **Then, we reweight these context-aware neurons of the LLM.**
>
> **3.**  **During online testing, we take the modified model for inference, without adding any inference time cost.**
>
> We suspect that such misunderstanding may be caused by the lack of clarity of the expression in Section 3.2: "**Ultimately, we allow each example to vote for the candidate neurons based on their attribution scores, and we select the top h neurons that receive the most votes.**". We will revise this sentence to "**Ultimately, we count the number of co-occurrences of neurons in all candidate sets, and we select the top h neurons with the highest number of co-occurrences as identified context-aware neurons.**" in the next version of our paper. Moreover, at the end of Section 3.2, we will add "These context-aware neurons are shared across all data instances." to enhance the clarity of the paper.
>
> **Re to Weaknesses #3: Safety issues raised by increasing faithfulness to the context.**
>
> Thank you for your insightful comments. Our research aims to address the scenario where the internal knowledge of LLMs is outdated or wrong, by enhancing the LLMs’ ability to process and capture the correct or domain-specific knowledge incorporated in the context. This capability is crucial in practical applications such as retrieval-augmented generation (RAG) and LLMs as agents.
>
> We acknowledge the existence of the scenario you mentioned. In this scenario, we can completely select to not adopt the proposed method and let the LLM use its own knowledge and ability to generate responses, by using a filter to identify jailbroken context.
>
> However, we firmly believe that effectively utilizing contextual knowledge and incorporating it into the generation process is an essential capability for LLMs. While not needed in every scenario, this ability is indispensable when required.

---

> > ### Comment · Reviewer_qHtz · 2024-08-13
> > **Thanks for the rebuttal!**
> >
> > Thanks for the responses, which addressed my questions. I will retain my positive scores.

---

> > > ### Author Response · Authors · 2024-08-14
> > >
> > > Thank you for your supportive comments and positive evaluations. We sincerely appreciate the time and effort you have dedicated to reviewing our responses.

---

### Author Rebuttal · Authors · 2024-08-07

Many thanks to all the reviewers for providing insightful comments and suggestions! We greatly appreciate you taking the time to review our work and provide constructive feedback to improve the quality of our paper.

The attached PDF shows all the comparison experiments we have done, including comparative experiments with Inference-Time Intervention (ITI) (Tables 1 and 2), with prompt engineering-based methods (Table 3), and with instruction-tuning (Table 4).

We will incorporate your comments and suggestions into the revised version of our paper.

---

### Decision · Program_Chairs · 2024-09-25

**Decision:**

Accept (poster)

**Comment:**

This paper aims to address the issues where LLMs’ parametric knowledge contradicts that in the context. A technique is proposed to mechanically identify neurons responsible for processing the context, after amplifying their influence on the model, the models are better at respecting the information in the context.

The reviews are unanimously positive. The response addressed most of the concerns, and authors already have a plan to incorporate the reviewers’ feedback into the revision. Accept.